# Structural Impact of Selected Retinoids on Model Photoreceptor Membranes

**DOI:** 10.3390/membranes13060575

**Published:** 2023-06-01

**Authors:** Szymon Radzin, Anna Wiśniewska-Becker, Michał Markiewicz, Sebastian Bętkowski, Justyna Furso, Joanna Waresiak, Jarosław Grolik, Tadeusz Sarna, Anna M. Pawlak

**Affiliations:** 1Department of Biophysics, Faculty of Biochemistry, Biophysics and Biotechnology, Jagiellonian University, 30-387 Krakow, Poland; 2Department of Computational Biophysics and Bioinformatics, Faculty of Biochemistry, Biophysics, Jagiellonian University, 30-387 Krakow, Poland; 3Department of Organic Chemistry, Faculty of Chemistry, Jagiellonian University, 30-387 Krakow, Poland

**Keywords:** photoreceptor membrane, retinoids, liposomes, spin labels, electron paramagnetic resonance

## Abstract

Photoreceptor membranes have a unique lipid composition. They contain a high level of polyunsaturated fatty acids including the most unsaturated fatty acid in nature, docosahexaenoic acid (22:6), and are enriched in phosphatidylethanolamines. The phospholipid composition and cholesterol content of the subcellular components of photoreceptor outer segments enables to divide photoreceptor membranes into three types: plasma membranes, young disc membranes, and old disc membranes. A high degree of lipid unsaturation, extended exposure to intensive irradiation, and high respiratory demands make these membranes sensitive to oxidative stress and lipid peroxidation. Moreover, all-trans retinal (AtRAL), which is a photoreactive product of visual pigment bleaching, accumulates transiently inside these membranes, where its concentration may reach a phototoxic level. An elevated concentration of AtRAL leads to accelerated formation and accumulation of bisretinoid condensation products such as A2E or AtRAL dimers. However, a possible structural impact of these retinoids on the photoreceptor-membrane properties has not yet been studied. In this work we focused just on this aspect. The changes induced by retinoids, although noticeable, seem not to be significant enough to be physiologically relevant. This is, however, an positive conclusion because it can be assumed that accumulation of AtRAL in photoreceptor membranes will not affect the transduction of visual signals and will not disturb the interaction of proteins engaged in this process.

## 1. Introduction

Vertebrate photoreceptor membranes contain a high level of polyunsaturated fatty acids (PUFAs) [1,2,3]. PUFAs amount to 27 up to 45% of total esterified fatty acids in photoreceptor outer segments’ (POS) phospholipids, among which is the most unsaturated fatty acid in nature, docosahexaenoic acid (22:6), amounting to 35% [1,2]. POS membranes are also enriched in phosphatidylethanolamines, in which mainly PUFAs are esterified [1,4]. The phospholipid composition and cholesterol content of subcellular components of photoreceptor outer segments enable to divide photoreceptor membranes into three types: plasma membranes (PM), young disc membranes (YDM), and old disc membranes (ODM) [5,6,7]. A high degree of lipid unsaturation [4], extended exposure to intensive irradiation [8,9], especially in the presence of endogenous sensitizers [10], and high respiratory demands [11] make these membranes sensitive to oxidative stress and lipid peroxidation [10].

The visual pigments, rhodopsin (Rh) in rods and three types of pigments in three types of human cones (S-, M-, and L-opsin), are also the main proteins present in the photoreceptor outer segments [12,13,14,15]. Such a high density of visual pigments, which reaches up to 20,000 molecules per μm^2^ of disc membrane in the case of Rh [16], allows effective absorption of light. Visual pigments belong to the family of opsins—transmembrane proteins with a characteristic structure, to which the photosensitive chromophore 11-*cis* retinal (11*c*RAL) is covalently bound via a protonated Schiff base bond [17]. 11*c*RAL, a vitamin A derivative, is conserved throughout evolution as a very efficient chromophore in most vertebrate visual pigments because of the high quantum yield of its isomerization reaction [18]. After absorption of the incident photon, 11cRAL in the chromophore-binding pocket of opsin isomerases into all-trans retinal (AtRAL) [18]. This ultrafast isomerization reaction is one of the fastest in nature [19]. AtRAL formed in photoactivated rhodopsin dissociates from the chromophore-binding site and is released into the photoreceptor plasma and disc membranes after the visual pigment’s deactivation [20]. AtRAL diffuses passively or is transported by the ATP-binding cassette transporter (ABCR) to the plasmatic site of a photoreceptor disc membrane [20], where it is enzymatically reduced to all-trans retinol (AtROL) by the NADPH-dependent all-trans-retinol dehydrogenase (RDH) [21]. The AtROL formed in this reaction is then transported to retinal pigment epithelial cells (RPE) [22]. Because the reduction of AtRAL to AtROL is quite slow, and therefore often considered as rate-limiting step in the visual cycle [23], it may lead to temporary accumulation of AtRAL in POS membranes. In special cases such as prolonged illumination leading to bleaching of almost 40% of the visual pigments or disturbances in its efficient transport, AtRAL released from rhodopsin may reach very high concentrations of up to 3 mM in the outer retina [10,24]. Such a high level of AtRAL, due to its photoreactivity and relatively high quantum yield of singlet oxygen photogeneration, may lead to retina photodamage [9,25,26]. Moreover, increased concentration of AtRAL in POS may lead to formation and accumulation of AtRAL conjugation products such as AtRAL dimer–phosphatidylethanolamine (AtRALdi–PE) or all-trans retinal dimers alone (AtRAL dimers) [25,27]. Additionally, some reports clearly suggest that both, AtRAL and its precursor 11cRAL, are substrates for bisretinoid formation in POS [28]. For a long time, it was believed that formation of these products requires light exposure to photoactivate rhodopsin, which then releases AtRAL. It has been shown, however, that RPE65−/− mice, which cannot produce 11cRAL, do not accumulate one of the well-known bisretinoids—A2E [29], a product of the reaction of two molecules of retinal and phosphatidylethanolamine [30]. It seems that AtRAL staying in POS prefers to bind to the phosphatidylethanolamine present in POS membranes forming an adduct called N-retinylidene-phosphatidylethanolamine (NRPE) [31,32]. NRPE is transported across photoreceptor disc membranes by the same transmembrane protein ABCR, enabling AtRAL to be reduced to AtROL by the cytoplasmic RDH following the dissociation of *N*-retinylidene-PE into retinal and PE [33]. Under some circumstances, which have not been fully elucidated, NRPE reacts with a second molecule of AtRAL instead of hydrolyzing to PE and AtRAL [32]. This reaction initiates a nonenzymatic synthetic pathway that leads to the formation of fluorescent di-retinal compounds within the lipid bilayers of POS including the abovementioned AtRALdi-PE, phosphatidylpyridinium bisretinoid (A2PE), and phosphatidyl-dihydropyridine bisretinoid (A2-DHP-PE) [32].

One of the di-retinal compounds, the AtRAL dimer, although less phototoxic than AtRAL alone [34,35], more easily undergoes photo-oxidation [36]. Products of its photodegradation may lead to formation of glyoxal and methylglyoxal, which in turn may cause irreversible modifications of retinal proteins [37]. It has also been shown that AtRAL dimers, if accumulated at a high level, contribute to retinal pigment epithelium cell degeneration [38]. It should also be remembered that the AtRAL dimer is a relatively large molecule with a wedge-like structure [39]. Therefore, it occupies more space between lipid molecules in the membrane and its possible structural impact on membrane properties should be taken into account.

All-trans retinol, as mentioned above, is a secondary product of visual-pigment bleaching, formed in the reaction of enzymatic reduction of AtRAL. In dark-adapted mice’s eyes its concentration reaches 36.0 ± 11.8 pmol per eye [40], while in photoreceptor outer segments its concentration has been assessed to reach 1.2 mM [41]. AtROL also appears to be the main source of fluorescence in POS, with a slow increase in intensity observed up to 40 min after rhodopsin deactivation [42]. Removing AtROL from POS is a rather complicated process and largely depends on light and the level of transport proteins [43].

Possible consequences of the accumulation of AtRAL and its conjugates in POS have been a subject of interest for several leading research groups in the world [24,32]. Their long-term studies, however, have been focused on possible phototoxic effects of these compounds on the function of photoreceptors and, especially, RPE cells. A potential structural impact of AtRAL, which accumulates in POS at such high concentration, on POS membrane properties, has not yet been studied.

In this work we focused just on this aspect. Biophysical properties of liposomes with lipid compositions resembling the lipid compositions of three types of photoreceptor membranes were studied employing electron paramagnetic resonance (EPR) spectroscopy and spin labeling technique in the absence and in the presence of all-trans retinal and other selected retinoids. Additionally, molecular dynamics (MD) simulations were used to confirm selected retinoids’ localization within the membrane.

## 2. Material and Methods

### 2.1. Chemicals and Lipids

1,2-dimyristoyl-sn-glycero-3-phosphocholine (DMPC), 1-palmitoyl-2-oleoyl-sn-glycero-3-phosphocholine (POPC), 1-palmitoyl-2-docosahexaenoyl-sn-glycero-3-phosphocholine (PDHAPC), 1-palmitoyl-2-oleoyl-sn-glycero-3-phosphoethanolamine (POPE), 1-palmitoyl-2-docosahexaenoyl-sn-glycero-3-phosphoethanolamine (PDHAPE), cholesterol (Ch), and the spin labels 1,2-dipalmitoyl-sn-glycero-3-phospho(tempo)choline (T-PC) and 1-palmitoyl-2-stearoyl-(n-doxyl)-sn-glycero-3-phosphocholine (n-PC, where n = 5, 7, 10, 12 or 16) were purchased from Merck KGaA (Darmstadt, Germany)

Retinoids: all-trans retinal (AtRAL) and all-trans retinol (AtROL) were purchased from Merck KGaA (Darmstadt, Germany). 11-cis retinal (11cRAL) was a generous gift from Lisa A. Neuhold, Program Director for Retinal Cell Biology and Development of Novel Therapies, National Eye Institute, NIH, USA. The chemical structures of the retinoids used in this work are shown in Figure 1.

### 2.2. All-Trans Retinal Dimer Synthesis

All-trans retinal dimer was synthesized and purified according to a previous report by Verdegem et al. [44]. AtRAL (1000 mg, 3.52 mM) was dissolved in 8 mL of anhydrous tetrahydrofuran (THF). NaH (60% dispersion in mineral oil) (162 mg, 4.04 mM) was washed with dry pentane in the glovebox and added to the reaction mixture (reactions carried out with neat 90% NaH gave 0% conversion). The mixture was stirred for 3 h at room temperature in the dark and the reaction was terminated by dropwise addition of a saturated ammonium chloride solution (NH_4_Cl). The mixture obtained was extracted with diethyl ether (3 × 20 mL). The organic phase was washed with brine and dried with anhydrous magnesium sulphate. The organic phase was evaporated under vacuum at room temperature. The AtRAL dimer was purified using silica gel column chromatography with n-hexane:Et_2_O (5:1). A total of 400 mg (41%) of the AtRAL dimer was yielded with 90% purity. The identity and purity of the AtRAL dimer was confirmed using ^1^H NMR spectroscopy, UV-VIS absorbance, and RP-HPLC with monitoring at 430 nm. The AtRAL dimer was lyophilized and stored at −80 °C under argon.

### 2.3. Preparation of Liposomes

According to the lipid composition of three types of membranes present in photoreceptor outer segments [5,6,7], liposomes with three different lipid compositions were prepared.

The plasma membrane model consisted of PC (DMPC, POPC) and PE (POPE, PDHAPE) in a 5:1 molar ratio with a low concentration (5 mol%) of docosahexaenoic acid residues esterified in PE. The cholesterol content was 40 mol%. Young disc model membranes consisted of PC (DMPC, POPC, PDHAPC) and PE (POPE, PDHAPE) in a 1:1 molar ratio with a high concentration (35 mol%) of docosahexaenoic acid residues esterified in both phospholipids and 30 mol% Ch. Old disc model membranes consisted of PC (DMPC, POPC, PDHAPC) and PE (POPE, PDHAPE) in a 1:1 molar ratio with a high concentration (35 mol%) of docosahexaenoic acid residues esterified in both phospholipids, and a low concentration of Ch, 5 mol%. Liposomes representing each type of photoreceptor membranes were enriched with selected retinoids (AtRAL, AtROL, 11-cis RAL, or AtRAL dimers). Liposomes not containing retinoids served as control samples. The lipid composition of the studied model membranes is presented in Table 1.

The liposomes were prepared using the following method. Briefly, chloroform solutions of lipids (containing 5 µmol of total lipids), ethanol solutions of respective retinoids (10 mol%, if applicable), and chloroform solutions of doxyl spin labels (1 mol%) were mixed, the organic solvent was evaporated with a stream of nitrogen, and the lipid film on the bottom of the test tube was additionally dried under reduced pressure (about 0.1 mm Hg) for 2–3 h. A volume of 0.5 mL of a buffer (PBS—phosphate buffered saline) was added to the dried film at a temperature well above the lipid phase transition temperature and vortexed vigorously. Then, the multilamellar liposome suspensions were subjected to freeze–thawing procedures, and centrifuged at 14,000× *g* for 10 min at 4 °C. The resulting pellet was then used for EPR measurements.

### 2.4. EPR Measurements

Due to their nitroxide-free radical moiety attached to the polar phosphatidylcholine headgroup or to the 5th, 10th, or 16th carbon atom in the acyl chain, applied phospholipid spin labels monitor different regions in the lipid bilayer. To get information about membrane fluidity, the order parameter S and correlation times *τ*_2*B*_ and *τ*_2*C*_ were calculated based on spin labels’ EPR spectra recorded at 310 K and 298 K. Representative EPR spectra of 16-PC and 5-PC with marked key values for calculation of these parameters are presented in Appendix A.

In the case of n-PC, the S parameter reflects the segmental order parameter of the hydrocarbon chain segment to which the nitroxide fragment is attached. This parameter is a measure of the semi-cone angle Θ_c_ within which the wobbling motion of this segment is confined [45]:S = cos Θ_c_ (1 + cos Θ_c_)/2 

The S parameter can be calculated based on the spectral parameters for all n-PC spin labels according to Marsh, 1981 [46]:S = 0.5407 (T′_II_ − T′_⊥_)/a_o_
a_o_ = (T′_II_ + 2T′_⊥_)/3
where 2T′_II_ and 2T′_⊥_ are the distances between the outer and inner spectral extrema, respectively, (Appendix A) and a_o_ is an isotropic hyperfine interaction constant.

The nearly isotropic motion of 16-PC allows employment of another approach to the interpretation of its spectra, namely use of the correlation times. They can be calculated according to two slightly different formulas, giving *τ*_2*B*_ and *τ*_2*C*_, respectively [47]:$$\tau_{2B}=6.51\cdot 10^{-10}\cdot ∆H_{0}\left[{\left(\frac{h_{0}}{h_{-}}\right)}^{\frac{1}{2}}-{\left(\frac{h_{0}}{h_{+}}\right)}^{\frac{1}{2}}\right]s$$
$$\tau_{2C}=6.51\cdot 10^{-10}\cdot ∆H_{0}\left[{\left(\frac{h_{0}}{h_{-}}\right)}^{1/2}+{\left(\frac{h_{0}}{h_{+}}\right)}^{1/2}-2\right]s$$
where *h*_0_, *h*_−_, and *h*_+_ are the amplitudes of the respective spectral lines, and Δ*H*_0_ is a linewidth of the central line (Appendix A). In the case of an isotropic movement, both times are the same or very similar. The more anisotropic the motion, the greater the difference between both times.

For polarity measurements, the z component of the hyperfine interaction tensor (2A_ZZ_) was obtained directly from the spectra of spin labels in frozen liposomes (134 K) as the distance between the outermost extrema. We related the 2Azz values in the membrane to those in the bulk solvent, by which the approximate dielectric constant ε at selected depths in the membrane can be estimated [48]. In detail, this method is based on the dependence of unpaired electron spin density at the nitrogen nucleus on solvent polarity. Polar solvents tend to increase the unpaired electron spin density at the nitrogen atom and, therefore, to affect the hyperfine interaction between the unpaired electron spin and the nitrogen nuclear spin. The enhanced interaction can be observed as an increase in 2Azz. Measuring 2Azz allows to distinguish between the motional and solvent effects on the spectrum because at low temperatures no appreciable molecular motion is detected at a time scale of 10^−7^ s. We have also shown in our old papers that the data obtained at low temperatures correlated well with measurements of ion penetration into various parts of the membrane performed at physiological temperatures [48,49].

The EPR measurements were performed using a Bruker EMX spectrometer (Bruker, BioSpin, Rheinstetten, Germany) equipped with a temperature control unit (EMX ER 4141 VT). The suspension of spin-labeled liposomes was placed in a gas-permeable capillary (i.d. 0.7 mm) made of TPX and located inside the EPR Dewar insert in the resonant cavity of the spectrometer. The sample was deoxygenated with nitrogen gas (about 10 min), which was also used for temperature control.

### 2.5. Statistical Analysis

Statistical analysis was performed using Sigma Plot 12.5 software (Alfasoft, Göteborg, Sweden). Analyzed data were obtained from three to four (in the case of all-trans retinol) independent measurements performed on new liposome batches. Statistically significant, marked with * or **, were the results for which the *p*-level was *p* < 0.05 and *p* < 0.02, respectively.

### 2.6. Molecular Dynamics Simulation

The four model systems used in this study consisted of 250 POPC, 16 selected retinoids, and 15,000 water molecules. A fifth system, without retinoid molecules, was used as a reference. In addition, sodium and chloride ions were added in physiologically relevant concentrations (0.15 M). The models of retinoid molecules were built using the Pymol program [50]. To diversify the location preference of AtRAL, 11*c*RAL, and AtRAL dimer molecules, eight retinoid molecules were placed in the hydrocarbon chain region of phospholipids, parallel to the POPC molecules: four retinoid molecules with the carbonyl group facing into the core of the membrane and four molecules with these groups facing into the water phase. The other eight were placed at the lipid–water interface, perpendicular to the normal orientation of the membrane surface. The CHARMM-GUI server [51] was used to build the bilayer system. For lipids and ions CHARMM36 parameters [52], for retinoids CHARMM/CGenFF parameters [53], and for water TIP3P parameters [54] were used. The particle mesh Ewald (PME) method [55] was used for computing Coulomb interactions. The non-bonded interactions were cut off at 12 Å. The 3D periodic boundary conditions with minimum image convention were used. All CH, CO, and OH bonds were constrained during simulations using the LINCS algorithm allowing the time step to be extended to 2 fs. Simulations were carried out under isothermal (310.15 K = 37 °C, which is above the main phase-transition temperature for a pure POPC bilayer (−5 °C) [56]) and isobaric (1 atm) conditions. The temperature was controlled independently for the solute and solvent using a Nosé–Hoover thermostat [57,58]. The pressure was controlled semi–isotropically using the Parrinello–Rahman method [59]. All four model systems were MD-simulated for 1 μs. After the first 300 ns of the simulation, all retinoid molecules had penetrated into the membrane. Therefore, the final 700 ns fragments of the MD trajectories were used for calculation of the average values. The orientational order (fluctuation) of POPC acyl chains was calculated using the molecular order parameter Smol. Smol for the nth segment of an acyl chain is defined through Smol = 0.5 · (3cos (θn) − 1), where θn is the instantaneous angle between the nth segmental vector (i.e., the (Cn − 1, Cn + 1) vector linking n − 1 and n + 1 carbon (C) atoms in the acyl chain) and the normal bilayer with corrections for double bonds [60,61]. The errors in average values of hydrogen bonds are standard deviation estimates, and errors in average values of Smol are standard errors calculated over 7 blocks, with each being a 100 ns fragment of trajectory.

## 3. Results and Discuss

In this work we studied the impact of selected retinoids: 11-cis retinal, all-trans retinal, all-trans retinol, and all-trans retinal dimers on biophysical properties of photoreceptor membranes in a model system. The cholesterol content and PC:PE ratio in the three types of liposomes used in these measurements resembled the lipid composition of three types of membranes naturally occurring in photoreceptor outer segments [62]. The impact of retinoids introduced into the studied model membranes at a concentration of 10 mol% with respect to lipids was investigated in terms of biophysical properties of these membranes, such as polarity, fluidity, and lipid order. Phosphatidylcholine ((18:0)(16:0)PC) with a nitroxyl radical moiety attached to the PC headgroup or selected carbon atoms in the stearyl chain introduced into studied liposomes at a concentration of 1 mol% enabled to employ EPR spectroscopy for this research.

The EPR spectra of nitroxyl labels introduced into the liposomes modeling PM, YDM, and ODM in the presence and in the absence (control) of selected retinoids were acquired at three different temperatures: physiological temperature (310 K), 293 K, and 134 K. The spectra of 5-, 10-, and 16-PC spin labels recorded in all types of studied model membranes at 310 K in the presence of AtRAL, 11cRAL, AtROL, and AtRAL dimers are shown in Figure 2. It can be seen that retinoids only slightly modified the EPR spectral parameters of nitroxyl radicals. However, the most different spectra compared to the control were acquired in the presence of 11cRAL and AtRAL dimers (Figure 2). This is probably due to the size and spatial orientation of these two molecules in the lipid bilayer. All the obtained results were divided and described in respect to the type of studied model membranes.

### 3.1. Plasma Membrane

The plasma membrane (PM) of photoreceptors contains two to three times more phosphatidylcholine than phosphatidylethanolamine [63,64], a high concentration of cholesterol, and a very low amount of polyunsaturated fatty acids [63,65].

As mentioned above, EPR spectra of nitroxyl radicals in all studied model membranes in the presence of retinoids were slightly modified (Figure 2). Small changes in EPR spectra were also observed at lower temperatures. The results of measurements performed at 134 K enabled to determine the local polarity at different depths of the studied membranes [48] and to plot the polarity profiles across the membranes in the absence and in the presence of different retinoids (Figure 3, Figure 5 and Figure 7). The polarity profiles of the model PM without retinoids (Figure 3, full symbols) correspond very well to our previously published data [66].

Introduction of 11cRAL and AtROL into the PM model significantly reduced membrane polarity, especially in the region close to its surface (Figure 3A,C). The observed effect was statistically significant (Appendix A) and persisted up to the depth of the 10th carbon atom in the lipid bilayer. The presence of AtRAL in turn induced only an insignificant decrease in polarity (Figure 3B) from the membrane surface to the 10th carbon atom and a slight increase in polarity in the center of the membrane. Statistical calculations showed that only the change inside the lipid bilayer was significant (Appendix A).

The decrease in membrane polarity induced by 11*c*RAL is very interesting. The 11*c*RAL molecule has one carbonyl group, which cannot participate in the formation of a hydrogen bond itself, although it can be its acceptor [67]. 11*c*RAL can therefore penetrate deeper into the membrane between lipid molecules, with the carbonyl group directed towards the surface of the membrane, but far enough from the area of active formation of hydrogen bonds. It can induce effective “closure” of polar lipid heads above each 11*c*RAL molecule while reducing the possibility of penetration of the upper areas of the membrane by water molecules [68]. The location and orientation of the functional groups of various molecules determine their ability to form hydrogen bonds with adjacent lipid and water molecules and thus determine the hydration state of the membrane surface. The existence of two differently oriented carbonyl groups has been observed in lipid monolayers on water, with the amount of “free” carbonyl groups, i.e., groups not forming hydrogen bonds, increasing with lipid density [69]. A similar effect may occur in membranes to which an additional pool of lipids (in this case retinoids) has been introduced. It can be expected that some 11cRAL molecules stay in the area of active hydrogen-bond formation and some go deeper toward the hydrophobic center of the membrane. The results of our MD simulation (see below) seem to confirm these two possible locations of 11*c*RAL.

The presence of 11*c*RAL in the PM-modeling liposomes probably also induces a slight increase in the volume per lipid molecule. This is due to the cis conformation of one (C11-C12) of the five unsaturated bonds in the 11*c*RAL structure (Figure 1), which bends the polyene chain of the molecule [70] at an angle of about 160 degrees [71]. Analysis of the non-planar spatial structure of the 11*c*RAL molecule shows that the three elements of the molecule (separated by the C6-C7 bond and the C12-C13 bond) lie in different planes relative to each other [72] increasing the space occupied by the molecule. Therefore, 11*c*RAL takes up more space in a membrane and disturbs the packing of lipid chains. Such an effect should be expected especially in the model membrane imitating the plasma membrane, which is characterized by a high concentration of Ch and a low content of PUFAs [73]. Interactions between straight and elongated chains of esterified saturated fatty acids in phosphatidylcholines, which are the main phospholipids building the PM, are stronger and the chains are more tightly packed than the chains of unsaturated lipids. In addition, a high content of Ch itself has an ordering effect on lipids and strengthens the interaction between them [74]. Thus, the introduction of a molecule with a bent polyene chain terminated with a ring additionally stabilized by the presence of methylene groups, into the layer of highly ordered lipids, disturbs their tight packing. Results of measurements of the lipid order parameter in the model of the PM, especially these obtained at 310 K, seem to confirm a slight disordering effect induced by 11*c*RAL in this membrane (Figure 4A).

The values of the order parameter S were the same in the presence and in the absence of AtRAL in all investigated positions within the membrane (Figure 4B). The lack of a significant effect of AtRAL on biophysical properties of the PM model membrane may be a consequence of its rapid reaction with phosphatidylethanolamine, which is present, although in a minor concentration, in this membrane. To investigate a possible and expected interaction of studied retinoids with cholesterol and PE, simple experiments were performed. In PM- and YDM-modeling liposomes, which contain large amounts of Ch, spontaneous lipid sorting and membrane-domain formation are highly probable [75]. 11cRAL, AtRAL, and AtROL were introduced into these types of liposomes, which were then incubated with triton X-100 on ice. This procedure leads to a separation of Ch- and saturated lipids-enriched domains (detergent-resistant membrane domains, DRM) and PUFA-enriched membrane domains (detergent-sensitive membrane domains, DSM) [76,77,78]. Lipids (including retinoids) present in both domains were extracted using Folch’s method [79], and the amount of each retinoid located in both domains was assessed using UV/vis spectroscopy (Appendix A). Collected absorption spectra are presented in Appendix A. The results of this experiment indicate that in PM-modeling liposomes 11*c*RAL localizes equally in both membrane domains (Appendix A). However, in YDM, in which the Ch concentration is similar to that PM but the PE concentration is much higher, 11*c*RAL localizes mainly in the PUFA-enriched membrane domain. Obviously, 11*c*RAL interacts with PE (which is mainly PUFA-esterified [80]) leading to formation of their reaction product with a slightly different absorption spectrum (Appendix A). The observed effect is even stronger in the case of AtRAL (Appendix A), as in this reaction all-trans retinal is favored over its cis-isomers [81]. Carbonyl groups of AtRAL in the presence of PE easily form Schiff bases with the amino groups of PE and such reaction products, although unstable and disappearing after dilution (Appendix A) preferentially stay in DSM (Appendix A). In DSM enriched in unsaturated lipids, a possible disturbing effect of the presence of retinoids is negligible. Spontaneous reactions of AtRAL with PE and the presence of its product in discs membranes has been well known for decades [1]. It is worth mentioning that AtRAL in liposomes containing various lipids, which spontaneously form different lipid domains in the membrane, but lacking phosphatidylethanolamines, localizes in DRM enriched in Ch and saturated lipids.

All double bonds present in the polyene chain of AtROL are in trans conformation, which makes it straight and elongated (Figure 1). On the other hand, the hydroxyl group of AtROL, which is more polar than the carbonyl group present in 11cRAL and AtRAL, places the AtROL molecule closer to the polar heads of membrane lipids [82]. A hydroxyl group itself can form hydrogen bonds with water molecules present at the membrane surface [83]. Due to the hydrophobic ring, which is probably located at the depth corresponding to the middle areas of esterified fatty acid chains in phospholipids, the effect of AtROL may resemble the effect of cholesterol. Cholesterol also contains a hydroxyl group, which anchors this molecule near the surface of the membrane and actively participates in the formation of hydrogen bonds. On the other hand, the steroid system of Ch interacts effectively with fatty acid residues, partially immobilizing them and causing an ordering effect [84]. AtROL, although to a lesser extent, clearly increases the order of lipids at a depth of the 10th carbon atom of fatty acid residues (Figure 4C). Consequently, it makes this part of the membrane less susceptible to deformation and protects the deeper layers of the membrane against penetration by water molecules.

### 3.2. Young Disc Membranes

Young disc (YD) membranes are characterized by comparable contents of PC and PE, as well as a high concentration of both PUFA-containing phospholipids and Ch. It seems that such a lipid composition minimizes the structural effects of retinoids, which were already rather weak in the PM model. Comparison of EPR spectra acquired in the absence (control) and in the presence of 11cRAL at 310 K showed almost undetectable differences between them (Figure 2B). The values of the 2A_ZZ_ parameter calculated from the spectra recorded at a temperature of 134 K indicated that the effect of the retinoids on the polarity of the YDM model was negligible, and the noted differences were not statistically significant (Appendix A). The plotted polarity profiles of YDM model in the absence and in the presence of three studied retinoids clearly confirmed these observations (Figure 5). Although the polarity of membranes containing 11cRAL and AtRAL in the region of the polar heads was lower than in the case of the control (Figure 5A,B), no visible differences existed in the deeper layers of the YDM model. It seems that due to the high content of PUFAs in the membranes of young discs, the introduction of a molecule with a bent polyene chain (11cRAL) or even a straight and elongated chain (AtRAL) is no longer of great importance, compared to the changes induced by the high degree of unsaturation of the lipids themselves [85]. The most significant, however, is the large amount of PE in YDM, which, as already discussed above, reacts efficiently with both isomers of retinal and determines the location of this reaction product in PUFA-enriched membrane domains. A reversed effect, though very weak, was observed when AtROL was present in the YDM model. In this case, the polarity slightly increased across the entire membrane (Figure 5C), which indicates the key role of the hydroxyl group of AtROL, which actively participates in the formation of hydrogen bonds and does not allow the AtROL molecule to enter deeper into the membrane [82]. An increased amount of PE, whose polar heads occupy a much smaller volume and adhere more closely to each other compared to phosphatidylcholines, leads to a self-assembly of PE molecules with a cone shape in the bilayer [86]. AtROL, which also localizes in DSM domains (Appendix A), enters between PE molecules pushing them apart and increasing water penetration into the membrane.

Most of the PUFAs present in photoreceptor outer segments are represented by docosahexaenoic acid (22:6, ω-3) with six double bonds in the cis conformation. This results in a much lower lipid packing in YDM compared to the PM [87], even in the presence of high concentrations of Ch. Thus, the addition of 11cRAL with its bent polyene chain to the membranes imitating young discs does not induce a visible effect on the degree of lipid packing (order parameter S) in the bilayer (Figure 6A). On the other hand, in case of AtRAL, which is equipped with a straight polyene chain terminated with a carbonyl group, a clear increase in lipid order parameter was noticeable (Figure 6B). It may be just the effect of enhancing of cholesterol ordering effect in this membrane or a result of specific, although unknown, orientation of the AtRAL–PE reaction product among lipid molecules. This assumption is supported by the fact that AtROL, which obviously does not react with PE, has a slightly reversed or no impact on lipid ordering in YDM.

### 3.3. Old Disc Membranes

Although the membranes of old discs (OD) in photoreceptor outer segments contain a comparable amount of PUFAs to that found in young discs, the concentration of Ch in these membranes is much lower [73]. This unique lipid composition makes these membranes very fluid and less ordered. Regeneration of rhodopsin in the OD membranes is much slower [88]; thus, temporary accumulation of AtRAL is significantly higher and formation of various condensation products (bisretinoids) is more probable. Therefore, the impact of all-trans retinal dimers, products of AtRAL conjugation, was additionally studied in the ODM model.

EPR spectra of nitroxyl labels in liposomes imitating the OD membrane containing AtROL and AtRAL dimers acquired at 310 K are presented in Figure 2C,D, respectively. None of the studied retinoids revealed a significant impact on the structural properties of the ODM model; however, AtRAL dimers seemed to be the most efficient in inducing noticeable changes in membrane polarity and lipid ordering. The calculated 2A_ZZ_ parameters for all studied retinoids in OD model membranes are presented in Appendix A. The polarity profiles across the ODM model in the absence and in the presence of all studied retinoids indicated that after the introduction of 11cRAL there was a slight increase in hydrophobicity in the region of the polar heads of the lipid bilayer (Figure 7A), but this change was not statistically significant (Appendix A). A reversed effect, i.e., a noticeable increase in membrane polarity in this region was caused by AtRAL (Figure 7B) and AtROL (Figure 7C), however to a lesser extent. High contents of PUFAs in this type of membrane accompanied by low concentrations of Ch cause loose packing of lipid molecules. Therefore, filling the space between lipid molecules with 11cRAL does not affect their interaction and degree of their ordering (Figure 8A). In the case of AtRAL however, a small increase in the lipid order parameter was observed (Figure 8B). This effect, observed already in YD model membranes, was clearly weaker in the case of ODM. It results from a much lower concentration of Ch, the impact of which seems to be strengthened by retinoids. A similar effect, already discussed in analogy to Ch action in membranes, should be expected for AtROL, but in this case detected differences in the S parameter in OD model membranes were negligible.

AtRAL dimers significantly increased the polarity inside the OD model membranes (Figure 7D). This molecule occupies a much bigger volume compared to the other retinoids and, entering between mainly loosely packed lipid molecules, makes them squeeze together to make enough room for it. The lipid order parameters determined at different depths of OD model membranes in the presence of AtRAL dimers confirmed this observation. It dramatically increased lipid ordering in OD model membranes between the 5th and 10th carbon atom of fatty acids chains inside the membrane.

### 3.4. Correlation Times

Due to the nearly isotropic motion of 16-PC in the membrane, parameters of its EPR spectra acquired at RT (293 K) and physiological temperature (310 K) may be used for determination of the spin label rotational correlation times, *τ*_2*B*_ and *τ*_2*C*_. The collected results obtained for all studied types of model membranes in the absence and in the presence of retinoids are presented in Figure 9. The *τ*_2*C*_ correlation time was clearly longer than the *τ*_2*B*_ time in all membrane models. However, the most significant difference between these times (up to 30%) occurred in the plasma membrane, while in the OD model membrane the observed difference did not exceed 10%, especially in samples measured at 310 K. These results confirm the significant effect of cholesterol on the anisotropy (freedom and direction) of possible movements of spin labels as well as the influence of Ch itself on the effect exerted by retinoids. The presence of both AtRAL and AtROL increased diffusion correlation times in all studied model membranes. However, in the case of AtRAL this effect was more pronounced (Figure 9) and was consistent with the observed increase in lipid ordering caused by AtRAL in the YDM model. Lipid mobility in the central region of the bilayer (depth of the 16th carbon atom in the chain) in all studied membrane models in the presence of 11cRAL was the same as in the control (Figure 9). This is because a bent polyene chain of 11cRAL makes this molecule shorter than its analogue AtRAL, in which all bonds are in the trans conformation. Although 11cRAL seemed to locate deeper into the lipid bilayer, its presence had no noticeable impact on 16-PC label surroundings. Significant impact, although only for the *τ*_2*C*_ time, was observed in OD model membranes in the presence of AtRAL dimers.

Summing up the obtained results, it seems that 11cRAL has very little effect on the biophysical parameters of young and old disc model membranes, i.e., in these membranes in which rhodopsin concentration is very high [89]. Although some structural changes due to elevated 11cRAL concentrations were observed in the PM model, it should be remembered that the concentration of 11cRAL in photoreceptor membranes never reaches a level close to 1mM as used in these studies. 11cRAL is introduced into photoreceptors to regenerate deactivated Rh, and this process seems to be quite well controlled [90]. Moreover, a mechanism exists in photoreceptor outer segments, which counteracts even the temporary accumulation of 11cRAL [90]. Thus, the structural effect of 11cRAL, even if well pronounced in the PM model, is unlikely to have physiological significance. Most important from a physiological point of view were studies concerning the role of AtRAL on structural properties of model membranes, as this retinoid accumulates in significant concentrations in POS membranes after Rh bleaching [24]. Our results clearly show that changes induced in these membranes by AtRAL are rather insignificant and do not have an impact on membrane function. The presence of AtRAL at a concentration reaching 10 mol% of the total concentration of lipids has a minor effect on membrane fluidity and polarity, properties which determine the rate of the first steps in the visual signal transduction process in photoreceptors.

### 3.5. Molecular Dynamic Simulation

All investigated retinoid molecules penetrated the membrane. Most molecules penetrated within the first 1–10 ns of the simulation, with only a few molecules remaining in the aqueous phase for the first 100–200 ns (Figure 10, Figure 11 and Figure 13). Z–coordinate plots for the center of mass and selected atoms of 16 molecules of 11cRAL, AtRAL, and AtRAL dimers used in MD simulations are presented in Appendix A. Immediate entry of the molecules, placed in the water phase above the POPC membrane at the start point of the simulation, into this membrane makes it difficult to pinpoint these molecules (Appendix A). Similarly, it is difficult to identify the retinoid molecules which were placed, at the beginning of the simulation, inside the membrane with their carbonyl groups directed to the interior of the membrane. Their rotation, directing the carbonyl group towards the surface of the membrane, happened very fast (Appendix A). During the simulation, virtually every molecule repeatedly passed from one layer of the membrane to another, with AtRAL molecules more often than AtRAL dimer ones, with the time scale of the process being 5–40 ns. All investigated retinoid molecules were located in the hydrophobic part of the membrane, with the retinoid carbonyl oxygen atoms at the height of the POPC carbonyl oxygen atoms and facing towards the interphase. For further analyses, the final 700 ns of the simulations, when all retinoid molecules were embedded inside the membrane, were considered. The results of performed MD simulations clearly indicated that in a simple POPC membrane, retinoid molecules locate close to the hydrophobic interior of the lipid bilayer, and it is there that a significant effect of the presence of retinoids should be expected. To confirm this observation experimentally, a series of EPR measurements was performed in POPC liposomes containing nitroxyl radicals in the absence and in the presence of AtRAL. Indeed, the results of this experiment showed that AtRAL dramatically increased the hydrophobicity of the POPC membrane interior. The polarity profile across the POPC membrane in the presence of AtRAL (Figure 12, open symbols) additionally confirmed how important for retinoid impact on membrane properties the other membrane components are, such as cholesterol and phosphatidylethanolamine. These lipids seem to determine retinoid location and behavior in the lipid bilayer.

The values of the order parameter (Smol) in POPC membrane alone and in the presence of 11cRAL, AtRAL, or AtRAL dimers were also calculated, based on MD simulations (Figure 14). The Smol parameter was calculated for palmitic acid chain esterified at the sn1 position (Figure 14A) and for oleic acid chain esterified at the sn2 position (Figure 14B) separately. In these chains all retinoids slightly increased the values of Smol, but the effect was stronger for the saturated chain. The ordering effect was, however, clearly more pronounced in the case of the AtRAL dimers. These results, performed on a simple unsaturated membrane model consisting of POPC, confirm the experimental data obtained from the EPR measurements, especially these performed on the ODM model (Figure 8D). The noticeable increase in lipid ordering in the ODM model was induced by AtRAL and, much stronger, by AtRAL dimers. In general, the Smol values obtained from MD simulations were lower than the S values obtained using EPR, because the simulated membrane did not contain cholesterol. Nevertheless, the results obtained using both techniques agreed very well showing a slight stiffening effect of AtRAL and AtRAL dimers on membranes, especially in their central part. An analysis of the number of hydrogen bonds formed between retinoids and water molecules was also performed (Table 2). Only a few hydrogen bonds with water were observed due to the location of the retinoid molecules in the POPC hydrocarbon chain region. AtRAL dimer molecules formed a lower number of hydrogen bonds, which may be due to the deeper localization of these molecules, resulting in their reduced exposure to the aqueous environment. Analysis of the effect of the presence of AtRAL and AtRAL dimer molecules on the number of hydrogen bonds between POPC molecules and water confirmed the negligible influence of retinoid molecules on the lipid–water interactions.

## 4. Conclusions

MD simulations performed for 11cRAL, AtRAL, and AtRAL dimers in simple POPC membranes indicate that the two former retinoids locate inside the membrane with the long axes of their molecules laying parallelly to lipid molecules, while the latter one penetrates even deeper into the membrane and localizes closer to the hydrophobic interior. All studied retinoid molecules align themselves with a carbonyl group directed towards the surface of the POPC membrane. Experiments performed in such simple POPC membranes containing AtRAL confirm that the most significant changes induced by AtRAL appear inside the lipid bilayer. However, in liposomes modeling three types of photoreceptor membranes, retinoids’ impact on their biophysical properties are determined and limited by the presence of cholesterol and phosphatidylethanolamine. Moreover, it seems that retinoids’ interaction with PE is more significant and stronger than their interaction with cholesterol. AtRAL and 11cRAL, especially in disc membranes, in which the PC to PE ratio is 1:1, react with PE leading to formation of a transient product via Schiff-base-type bonds.

The highly positive outcome of our study is that the temporary accumulation of all-trans retinal in photoreceptor outer segment membranes of all types has no significant impact of biophysical properties of these membranes. Their unique lipid composition, i.e., the presence of a high concentration of cholesterol in the case of PM and YD membranes, especially a high level of PE in young and old disc membranes, prevents them from dramatic changes which could be caused by retinoids, especially AtRAL, which accumulates there in significant concentrations.

## Figures and Tables

**Figure 1 membranes-13-00575-f001:**
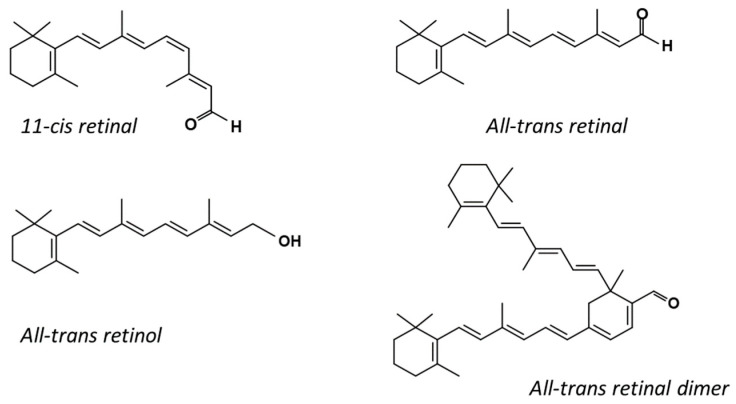
Chemical structures of selected retinoids.

**Figure 2 membranes-13-00575-f002:**
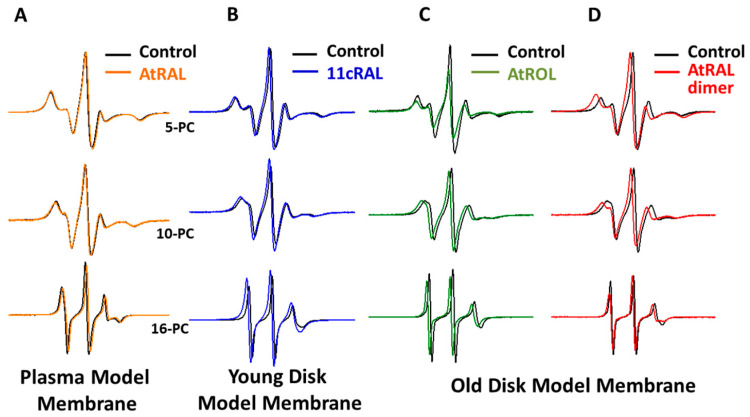
Representative EPR spectra of 5-PC, 10-PC, and 16-PC nitroxyl labels introduced into liposomes modeling photoreceptor plasma membranes (**A**), young disc membranes (**B**), and old disc membranes (**C**,**D**). Spectra acquired in liposomes without retinoids (**A**–**D**, control, black lines) and in the same type of model membrane liposomes containing 10 mol% of all-trans retinal (**A**, orange line), 11-cis retinal (**B**, blue line), all-trans retinol (**C**, green line), and all-trans retinal dimers (**D**, red line). Spectra were acquired at 310 K in the absence of oxygen.

**Figure 3 membranes-13-00575-f003:**
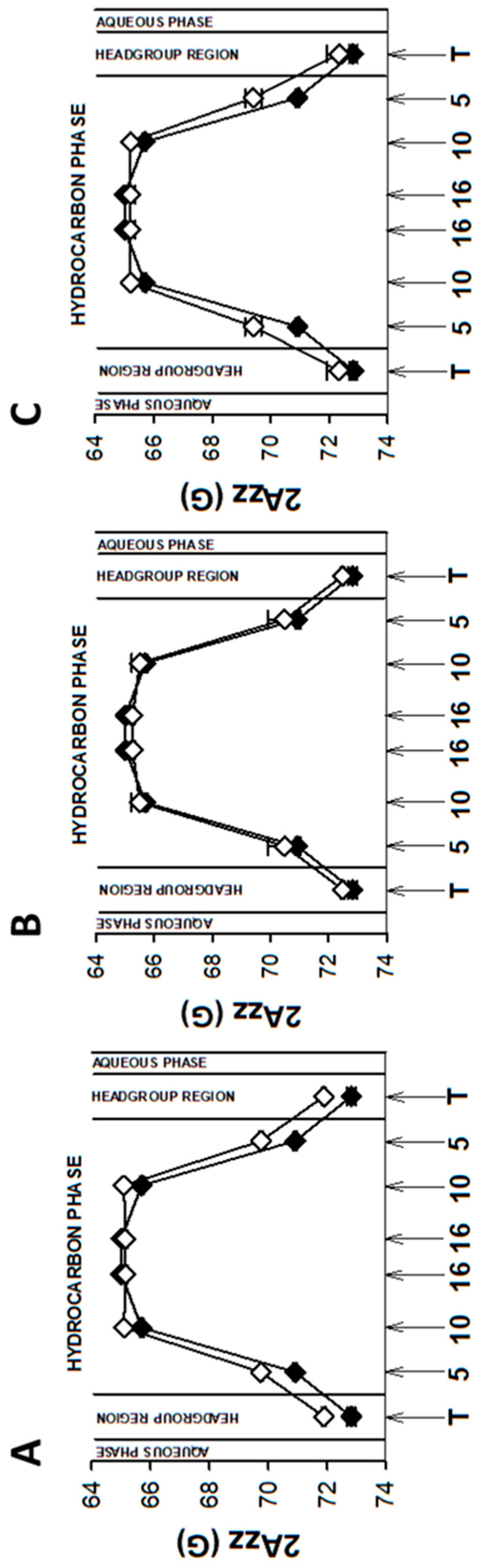
Polarity profiles across photoreceptor plasma membrane models in the absence and in the presence of selected retinoids: 11-cis retinal (**A**), all-trans retinal (**B**), and all-trans retinol (**C**). The spectra were acquired at 134 K. Upward changes in 2Azz indicate a decrease in polarity. Full symbols denote control membranes and empty symbols membranes containing retinoids. Approximate locations of the nitroxide moieties of spin labels are indicated by arrows.

**Figure 4 membranes-13-00575-f004:**
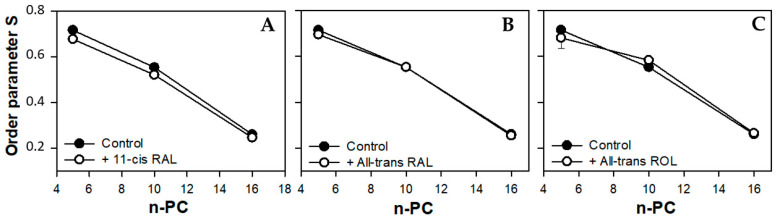
Values of the order parameter S obtained from the spectra of 5-, 10-, and 16-PC spin labels in the model of the plasma membrane made of synthetic lipids in the absence (full circles) or in the presence (open circles) of selected retinoids: 11-cis retinal (**A**), all-trans retinal (**B**), and all-trans retinol (**C**). The spectra were acquired at 310 K.

**Figure 5 membranes-13-00575-f005:**
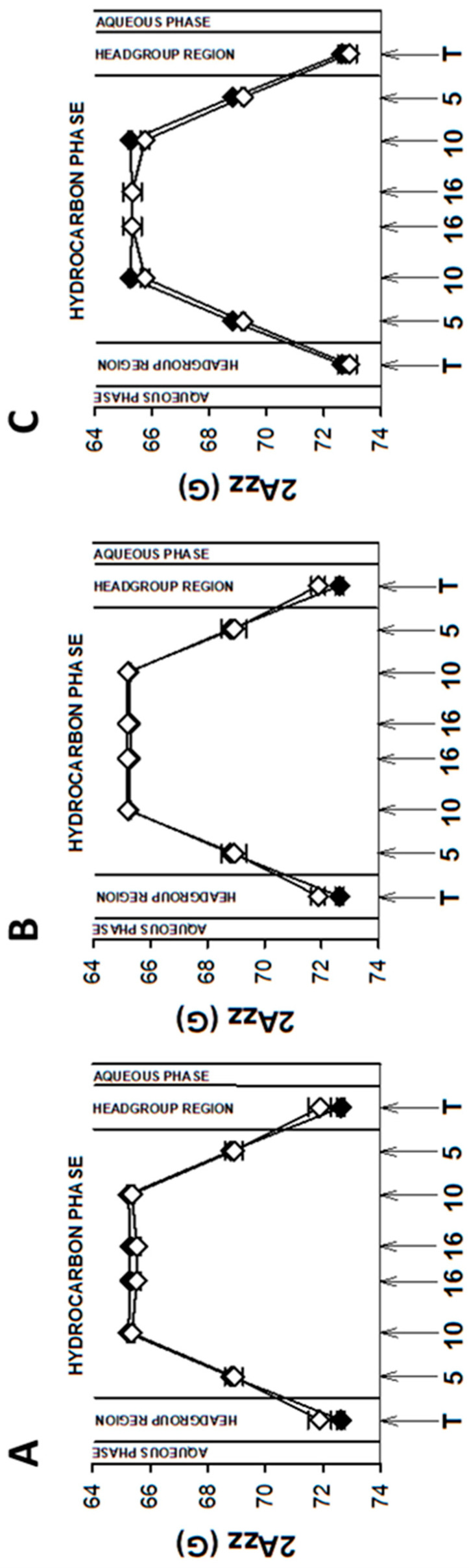
Polarity profiles across young disc membrane models (PC:PE, 1:1; Ch, 30 mol%; PUFAs, 35 mol%) containing 11-cis retinal (**A**), all-trans retinal (**B**), and all-trans retinol (**C**). The spectra were acquired at 134 K. Upward changes in 2Azz indicate a decrease in polarity. Full symbols denote control membranes and empty symbols membranes containing retinoids. Approximate locations of the nitroxide moieties of spin labels are indicated by arrows.

**Figure 6 membranes-13-00575-f006:**
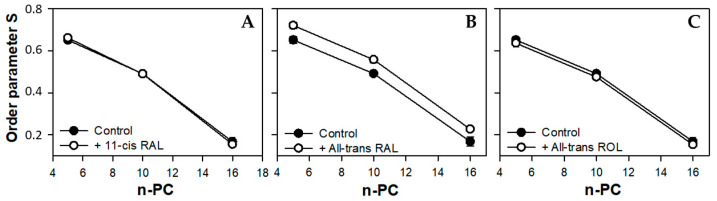
Values of order the parameter S obtained from the spectra of 5-, 10-, and 16-PC spin labels in the model of young disc membranes made of synthetic lipids in the absence (full circles) or in the presence (open circles) of selected retinoids: 11-cis retinal (**A**), all-trans retinal (**B**), and all-trans retinol (**C**). The spectra were acquired at 310 K.

**Figure 7 membranes-13-00575-f007:**
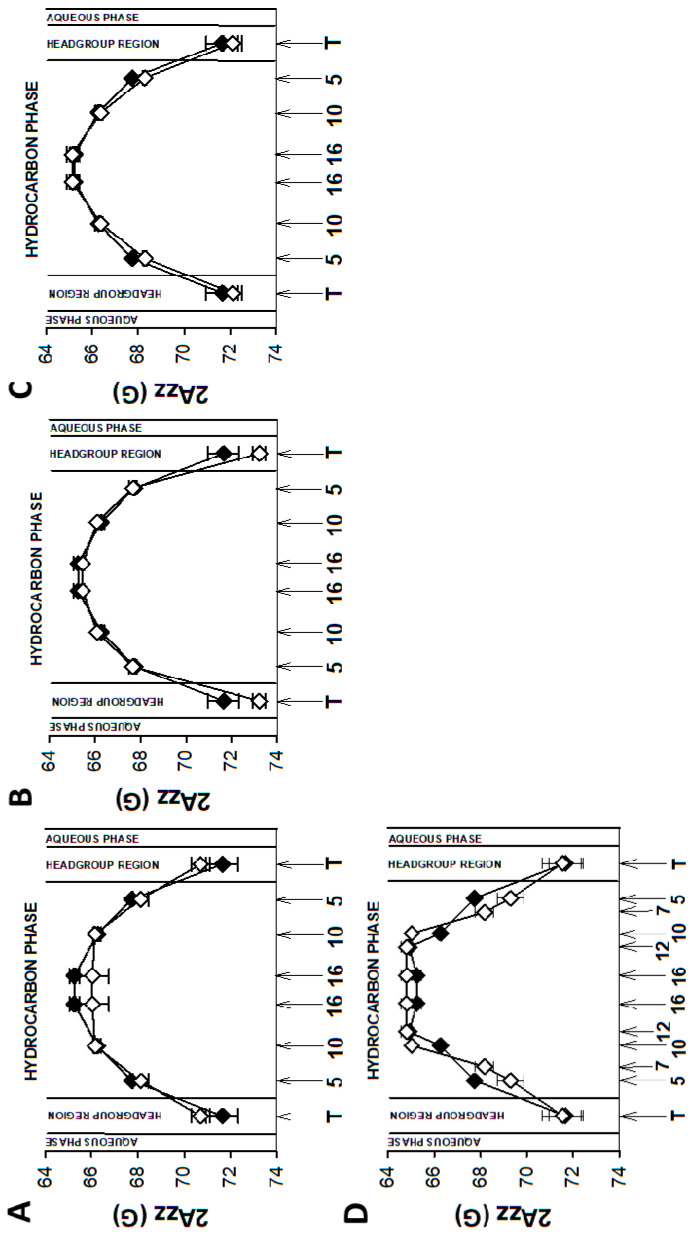
Polarity profiles across old disc membrane models (PC:PE, 1:1; Ch, 5 mol%; PUFAs, 35 mol%) containing 11-cis retinal (**A**), all-trans retinal (**B**), all-trans retinol (**C**), and all-trans retinal dimers (**D**). The spectra were acquired at 134 K. Upward changes in 2Azz indicate a decrease in polarity. Full symbols denote control membranes and empty symbols membranes containing retinoids. Approximate locations of the nitroxide moieties of spin labels are indicated by arrows.

**Figure 8 membranes-13-00575-f008:**
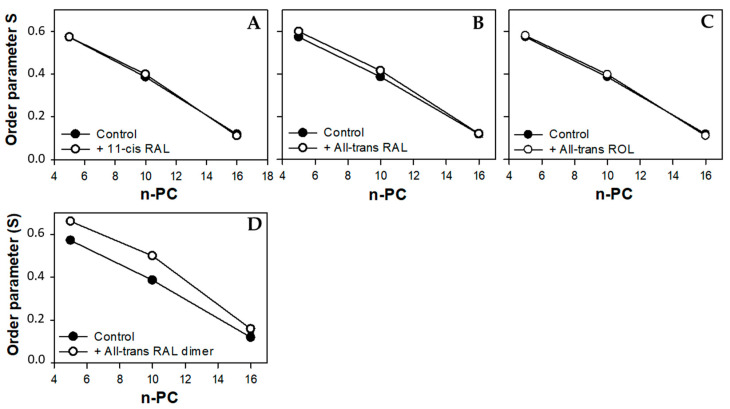
Values of the order parameter S obtained from the spectra of 5-, 10-, and 16-PC spin labels in the model of old disc membranes made of synthetic lipids in the absence (full circles) or in the presence (open circles) of selected retinoids: 11-cis retinal (**A**), all-trans retinal (**B**), all-trans retinol (**C**), and all-trans retinal dimers (**D**). The spectra were acquired at 310 K.

**Figure 9 membranes-13-00575-f009:**
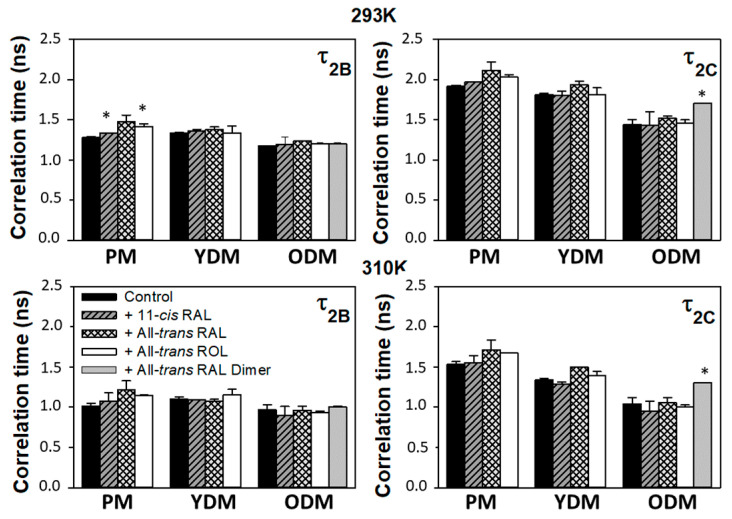
Rotational correlation times *τ*_2*B*_ and *τ*_2*C*_ (ns) of 16-PC nitroxyl radicals in liposomes modeling three types of photoreceptor outer segment membranes: the plasma membrane (PM), young disc membranes (YDM), and old disc membranes (ODM) in the absence (control) and in the presence of studied retinoids: 11-cis retinal (11-cis RAL), all-trans retinal (All-trans RAL), all-trans retinol (All-trans ROL), and all-trans retinal dimers (All-trans RAL dimer). The legend is valid for all charts. Measurements were performed at 293 K (upper charts) and 310 K (lower charts). Statistically significant changes in rotational correlation times of 16-PC nitroxyl radical in studied liposomes induced by retinoids has been labelled with asterix (*p* < 0.05).

**Figure 10 membranes-13-00575-f010:**
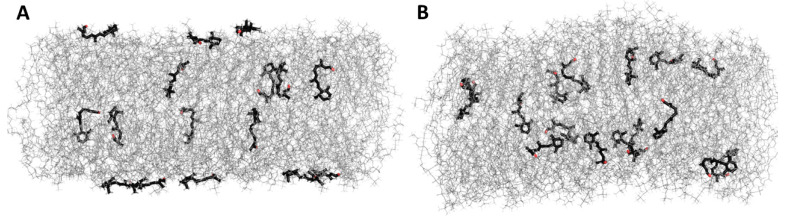
MD simulation of 11-cis retinal in the POPC membrane. (**A**)—start, 0 ns, (**B**)—end, 1 μs. Water molecules are removed for clarity.

**Figure 11 membranes-13-00575-f011:**
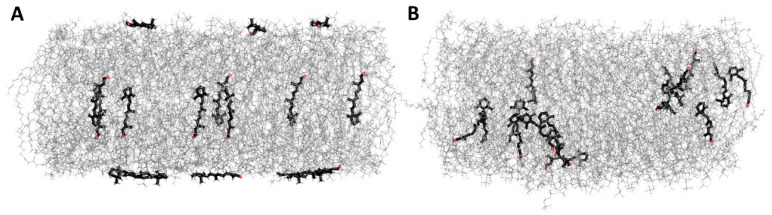
MD simulation of all-trans retinal in the POPC membrane. (**A**)—start, 0 ns, (**B**)—end, 1 μs. Water molecules are removed for clarity.

**Figure 12 membranes-13-00575-f012:**
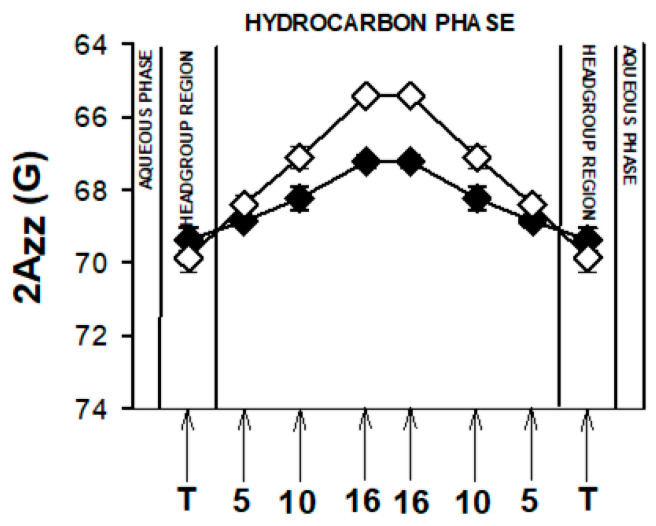
Polarity profiles across POPC membranes containing all-trans retinal. The spectra were acquired at 134 K. Upward changes in 2Azz indicate a decrease in polarity. Full symbols denote control membranes and empty symbols membranes containing AtRAL. Approximate locations of the nitroxide moieties of spin labels are indicated by arrows.

**Figure 13 membranes-13-00575-f013:**
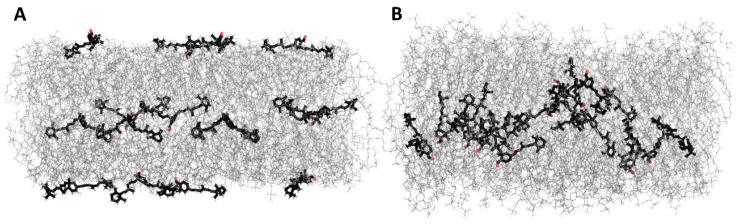
MD simulation of all-trans retinal dimers in the POPC membrane. (**A**)—start, 0 ns, (**B**)—end, 1 μs. Water molecules are removed for clarity.

**Figure 14 membranes-13-00575-f014:**
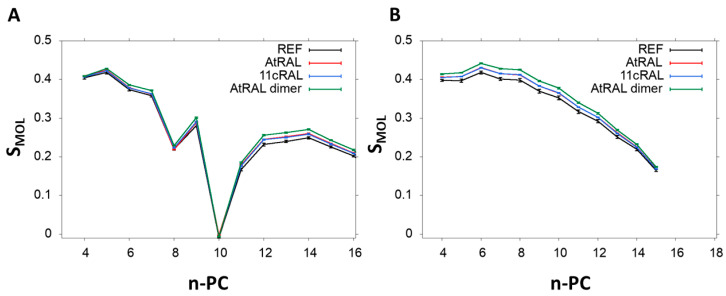
Molecular order parameter (Smol) calculated for POPC alone (REF) and in the presence of all-trans retinal or all-trans retinal dimers plotted against the carbon number in the alkyl chain. The Smol parameter was calculated for palmitic acid chain esterified at the sn1 position (**A**) and for oleic acid chain esterified at the sn2 position (**B**) separately.

**Table 1 membranes-13-00575-t001:** Lipid composition of studied model membranes resembling three types of membranes naturally occurring in photoreceptor outer segments.

	Plasma Membrane	Young Disc Membrane	Old Disc Membrane
PC:PE ratio PUFA content	PC:PE, 5:1 PUFA 5 mol%	PC:PE, 1:1 PUFA 35 mol%	PC:PE, 1:1 PUFA 35 mol%
DMPC	2 mM	1 mM	1 mM
POPC	3 mM	1.5 mM	2.75 mM
(16:0)(22:6)PC	-	1 mM	1. mM
(16:0)(22:6)PE	0.5 mM	2.5 mM	2.5 mM
POPE	0.5 mM	1 mM	2.25 mM
Cholesterol	4 mM	3 mM	0.5 mM
n-PC	0.1 mM	0.1 mM	0.1 mM
Retinoid	0 mM (control) or 1 mM	0 mM (control) or 1 mM	0 mM (control) or 1 mM

**Table 2 membranes-13-00575-t002:** Average number of hydrogen bonds between retinoid molecule (considering single atoms) and POPC (considering oxygen atoms of phosphate (Op) and carbonyl groups (Oc)) and between retinoid and water (RET-WATER). POPC: Op—phosphate group, Oc—carbonyl group. Only retinoid molecules which are present in the membrane are considered.

	POPC-WATER	POPC Op-WATER	POPC Oc-WATER	RET-WATER
REF	7.07 ± 0.1	5.27 ± 0.07	1.8 ± 0.06	———
11-cis RAL	7.14 ± 0.1	5.32 ± 0.07	1.83 ± 0.06	0.41 ± 0.15
All-trans RAL	7.13 ± 0.1	5.31 ± 0.07	1.82 ± 0.06	0.63 ± 0.16
All-trans RAL DIMER	7.15 ± 0.1	5.32 ± 0.07	1.83 ± 0.06	0.41 ± 0.14

## Data Availability

Data created during implementation of this project will be deposited and then available at Jagiellonian University Repository.

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
