# Peer review of "Structural Impact of Selected Retinoids on Model Photoreceptor Membranes"

_membranes, 2023, doi:10.3390/membranes13060575_

Round 1
Reviewer 1 Report
The manuscript of Radzin et al. investigates the impact of retinoids on biophysical properties of model membranes, the lipidic composition of which resemble the ones of membranes photoreceptors. In particular, they investigate the impact of certain retinals, which are the co-factors that enable for light sensing, on the polarity and the order parameter of the model membranes. To this end, the authors add spin-labels at different carbon atoms of the lipids’ fatty acids and employ electron spin resonance (ESR) to determine the above-mentioned parameters. The experimental investigations are complemented by molecular dynamics (MD) simulations.
In general, the authors touch an interesting and not well-addressed topic. The manuscript is mostly well-written but far too long, especially as the addition of retinoids introduces only very small, and typically (statically) insignificant changes to the membrane properties. As a consequence, the sections on the different retinoids appear quite repetitive. Nevertheless, even the absence of notable changes is, in principle, an interesting observation and justifies publication, provided that the manuscript is restructured to remove the redundancies.
MAJOR ISSUES
1. I don’t see the point to present the Azz data as figures and in tables. I would keep the figures and shift the tables to the supplementary information (or vice versa). Similarly, the effects seem to be very similar, irrespective if the membranes mimics the plasma membrane or the so-called Young and Old membrane. Maybe the authors can cut down here (by highlighting the similarities and only discussing the differences). In addition, the figures have a quite low resolution currently, which has to be improved.
2. The authors indicated significant changes by * and **. Maybe I missed it, but did they specify somewhere the significance test used and to which p-levels * and ** correspond to? How many experiments enter into the calculation of the mean value and confidence interval? Are these independent measurements (using new liposome batches) or repetitions of the same batch?
3. In the Azz data, significant changes in comparison to the control are typically observed close to the surface. The MD simulations, however, show that the retinoids accumulate in the middle of the membranes. Hence, I would have expected to also see most changes in the middle and not at the surface of the membrane. Can the authors comment on this?
4. Why was cholesterol not included in the MD simulations? If retinoids can be added, why not cholesterol?
5. Page 5, line 212: “Membrane polarity” Can the authors define, which property they refer to? Polarity typically implies an asymmetry across the membrane (e.g., differences in membrane composition of the inner and outer leaflet). Or is kind of a solvent quality referred to (polar vs. non-polar environment)?
6. Furthermore, the “polarity” measurements have been done at 134 K. How can the authors be sure to have the same membrane structure in comparison to 310 K (i.e., can phase transitions be ruled out)?
7. Page 6, lines 217 – 221 and lines 228 - 229: Are these arguments based on experimental data or assumptions?
8. Page 6, lines 235 – 236: “Our unpublished results …” It is difficult to access conclusion, which are based on data that is not enclosed in manuscript or published elsewhere.
9. Page 6, lines 241 – 242: “… seem to confirm slight disordering effect induced by 11cRAL in this membrane (Fig. 4A)” I don’t see a clear trend with respect to ordering and disordering for the different retinoids. Is any of them statistically significant? Furthermore, the MD simulations suggest that if there is an effect, it will be very small in comparison to the measurement accuracy. Can the authors comment on this?
10. Page 13, lines 426 – 429: “… effect of cholesterol on anisotropy” How can anisotropy (freedom and direction) be assessed in the experimental setting? As far as I understand, ESR probes spin resonance at different depths of the membrane? But how does it provide information on the direction of the spin labels in space?
Author Response
Dear Sir or Madam,
All sections of the Manuscript have been improved according to suggestions of both reviewers. Introduction has been extended with additional information on the role of retinoids and the visual cycle. In consequence, the references list became much longer, unfortunately it was inevitable. A detailed description of employed methods and way of calculation of presented structural parameters have been added to the Material and Methods section. To complete the work, supplementary materials were prepared.
Detailed response to Reviewer 1 has been upload as a pdf file.
We believe that the additional experiments we performed and the newly obtained results we introduced, made our work more valuable and suitable for publication in Membranes.
Sincerely Yours,
Anna Pawlak

Reviewer 2 Report
In the introduction the link between the biological aspects of photodetectors and the lipid membrane organization is not clear, neither the motivation for purely biophysical studies of model membranes (authors just mention it has never been done)
The reference on photoreceptor membrane models should be added in the introduction and materials and methods
The exact mole fractions of each lipid component in the mixtures should be given in such a way that the sum yields 1
Why molecular simulations were performed only with POPC?
The authors should justify the selected concentration of retinoids (10% mol of the total lipid)
The quality of Figures 3, 6 and 9 is very poor and makes the figures difficult to be read
The results section is quite heavy to read, it should be separated by more paragraphs with corresponding figures in between to make the reading more fluid
Tables 2 to 5. There are too many digits in the values and related uncertainties, please correct that
A detailed description of how to calculate order parameters using the different spin labels is missing
The authors mention at some point: 'Our unpublished results also indicate that 11cRAL preferentially localizes in membrane domains enriched in Ch and saturated lipids' How can the authors prove this sentence ? They should mention techniques that could help resolving these domains such as AFM and the possibility of checking other properties such as possible changes in the nanomechanics of bilayers upon the presence of these retinoids. They can cite a reference review on how those measurements could be performed: L. Redondo-Morata, P. Losada-Pérez, M.I. Giannotti, Lipid bilayers: Phase behavior and nanomechanics, Current topics in membranes 86, 1 (2020).
At some point, the authors mention the volume occupied by a lipid molecule. Could this property be estimated by density measurements for the lipid mixtures under study?
How does the distribution of retinoids in cholesterol-rich possible domains affect the rotational diffusion changes?
What is the difference between Figures 13 and 14? This is not clear at all
The conclusions are quite short compared to the length of the paper. The authors should elaborate more on the main findings, even if changes are not that significant and what they imply, and highlight the effect of cholesterol in plasma membranes in a clearer way. They might mention the needs of more local techniques, like AFM to actually visualize or quantify changes in other properties than the ones they evaluated due to the presence of these retinoids
Author Response

(The authors gave the same response as above.)

Round 2
Reviewer 2 Report
Publish as it is